# Determinants for Drunk Driving Recidivism—An Application of the Integrated Prototype Willingness Model

**DOI:** 10.3390/bs15010048

**Published:** 2025-01-05

**Authors:** Rong-Chang Jou, Han-Wen Hsu

**Affiliations:** Department of Civil Engineering, National Chi Nan University, Nantou 54561, Taiwan; rcjou@ncnu.edu.tw

**Keywords:** prototype willingness model, recidivism, drunk driving

## Abstract

The paper applies the prototype willingness model (PWM) and incorporates components of the theory of planned behavior (TPB), along with deterrence factors, to understand the behavioral intentions, willingness, and recidivism behaviors of individuals penalized for drunk driving. It explores psychological and social factors influencing repeat offenses, focusing on attitudes, subjective norms, prototypes, and deterrence. The PWM outlines two pathways—reasoned (based on intentions) and social reactive (based on willingness). The model helps predict risky behaviors like drunk driving. Thirteen hypotheses are proposed in this study to examine how various factors, such as attitudes, subjective norms, and deterrence, influence willingness, intentions, and behavior. Surveys were conducted among individuals attending road safety classes after being penalized for drunk driving. A total of 1156 individuals participated in the survey, with 855 valid responses collected. The results indicate that behavioral willingness had a stronger impact on recidivism than intention. On the other hand, subjective norms did not significantly affect the intent to reoffend, but attitudes, deterrence, and PBC did. The findings suggest that focusing on behavioral willingness, deterrence, and educational interventions could help reduce repeat drunk driving offenses. The paper offers insights for policymakers to improve prevention strategies, by focusing on the psychological motivators of repeat offenders.

## 1. Introduction

Drunk driving is a significant global social issue. It is estimated that 400 million people worldwide, or 7% of the population aged 15 and older, suffer from alcohol use disorders ([69]). Among them, 209 million people (3.7% of the global adult population) are affected by alcohol dependence ([69]). Alcohol consumption not only harms the individual who drinks, but also poses significant risks to others. A substantial portion of the disease burden attributed to alcohol comes from injuries, particularly road traffic crashes. In 2019, out of 298,000 deaths caused by alcohol-related road crashes, 156,000 were the result of someone else’s drinking ([69]). The National Highway Traffic Safety Administration (NHTSA) estimates that about 37 people in the United States die in drunk driving crashes each day—that is one person every 39 min ([51]). In 2022 alone, 13,524 people lost their lives in alcohol-impaired driving crashes ([51]). The most effective penalties to reduce alcohol-induced fatal crashes are zero-tolerance alcohol use restrictions on underage drivers, prohibiting open alcoholic beverages in motor vehicles, and laws that restrict vehicle use among past DUI (driving under the influence) perpetrators, namely license suspensions and mandating ignition interlock devices ([70]).

Research shows that binge drinking increases road crashes by 18.6%, fatal road crashes by 72%, injury-related accident and emergency (A&E) attendances by 6.6%, and the number of arrests related to alcohol incidents by 71% ([29]). To enhance the effectiveness of drunk driving laws, it is necessary to increase the certainty of punishment, establish and maintain social norms against driving under the influence, and implement additional effective measures to reduce recidivism among drunk drivers ([22]). In China, the average annual incidence of traffic crashes, mortality, and injury rates decreased after the implementation of drunk driving laws ([24]). A study estimated that the annual cost attributable to alcohol exceeds USD 165 million, further calculating that alcohol-related hospital and outpatient expenses amount to USD 128.21 million ([14]). This highlights the significant burden of alcohol on public health systems. Therefore, alcohol use continues to impose a significant economic burden on modern societies. If all alcohol-related harms are included, the costs are estimated to average 1306 international dollars per adult, accounting for approximately 2.6% of the total GDP in the countries studied ([50]).

In Taiwan, drivers of motor vehicles who have a breath alcohol concentration of 0.25 milligrams per liter or a blood alcohol concentration of 0.05% or higher are subject to a penalty of imprisonment for up to 3 years, with the possibility of a fine not exceeding TWD 300,000 (1 USD = 30 TWD). The Taiwanese police established temporary drunk driving checkpoints on major roads to randomly inspect suspicious vehicles. Offenders are immediately tested for alcohol concentration, and penalized based on the results. Government agencies also use media campaigns and organize traffic safety-related courses to raise awareness about the dangers of drunk driving. However, for individuals who commit a third drunk driving offense within five years, regardless of whether the offense results in injury or death, they are not eligible for a fine, but must serve a prison sentence.

According to the data compiled in this study on the number of fatalities in A1-category road traffic crashes in Taiwan “https://www.npa.gov.tw/ch/app/folder/592 (accessed on 26 December 2024)” from 2009 to 2024 (Summarized in Table 1), the number of deaths caused by drunk driving in A1-category crashes ranged between 376 and 439 people from 2009 to 2012. In 2013, legal amendments were enacted to establish clear legal standards for impaired driving due to alcohol consumption, and to increase the minimum statutory penalties, stipulating that offenders may face imprisonment of up to two years and may also be subject to a fine of up to TWD 200,000. As a result, the number of deaths caused by drunk driving in A1-category crashes decreased to 245 in 2013, and by 2023, the number of fatalities caused by drunk driving in A1-category road traffic crashes had further dropped, to 137.

Based on current trends, increasing the penalties for drunk driving has, indeed, demonstrated a deterrent effect, as evidenced by the decrease in related incidents. This study seeks to further explore the psychological aspects of drunk drivers, to gain a deeper understanding of their mental and cognitive characteristics. By analyzing how various factors influence behavioral intentions and willingness of drunk driving, the study examines the impact these have on recidivism. Additionally, relevant dimensions are introduced to conduct an exploratory analysis of the overall model. Investigating the intentions, willingness, and recidivism behaviors of drunk driving offenders is crucial for the prevention and control of drunk driving, as these offenders represent a high-risk group for addressing this issue. Gaining a deeper understanding of their behaviors can help formulate more effective prevention and intervention strategies. By analyzing the behavioral patterns and psychological motivations of repeat offenders, policymakers can develop targeted legal frameworks, educational programs, and social counseling measures to reduce the rate of repeat offenses and improve traffic safety. Furthermore, exploring the relationship between drunk driving intentions and recidivism can assist governments in establishing early prevention mechanisms, thereby reducing the social costs and casualties caused by drunk driving.

Research shows that decision-making is often influenced by context, with individuals frequently making reactive choices, based on their immediate situation ([34]); ([36]; [55]). The prototype willingness model explains this reactive process, by highlighting how risky behavior can arise from unintentional willingness and social influences ([33]; [34]; [36]). As a result, the PWM was developed to explain risky behaviors among adolescents, such as smoking, binge drinking, and reckless driving ([35]). PWM focuses on understanding the psychological factors that determine a person’s willingness to engage in specific behaviors ([37]). In recent years, the PWM has been widely used in traffic psychology to study various behaviors, including drunk driving ([71]), speeding ([59]), adolescent aggressive riding behavior ([77]), risky driving behavior ([68]), pedestrian violations ([15]) and red-light running behavior ([21]; [66]). In this model, a prototype represents the social image of a typical person engaging in a particular behavior ([37]), and its aim is to identify the psychological predictors that influence an individual’s willingness to engage in such behaviors ([37]).

Recidivism of drunk driving is a global social issue. Research shows that recidivism rates for drunk driving remain between 21% and 47% within five years ([26]; [52]). Drunk driving recidivists are also 62% more likely to be involved in fatal crashes compared to other drivers ([25]). Additionally, the risk of traffic crashes for drunk driving recidivists is higher than that of first-time offenders ([17]). In Taiwan, drunk driving is classified as a criminal offense, and over one-third of offenders have previous drunk driving violations on record ([74]).

Therefore, this study investigates the psychological aspects of individuals who engage in drunk driving through the application of the prototype willingness model. A survey was conducted with participants enrolled in traffic safety courses due to drunk driving offenses, aiming to analyze their psychological and cognitive characteristics. The research examines how various factors influence both the behavioral intentions and willingness of drunk drivers, specifically focusing on the role of willingness in shaping intentions and the likelihood of repeat offenses. Furthermore, it explores the effects of deterrence factors and perceived behavioral control on these drivers’ behavioral intentions, willingness, and actual repeat behavior. The ultimate objective is to develop a comprehensive model to inform future research and interventions, aimed at preventing repeat drunk driving offenses.

The primary contribution of this study lies in the systematic investigation of the recidivism intentions, willingness, and behaviors of individuals with prior drunk driving experiences, thereby addressing a gap in the current research on this specific group. At the national level, such a survey provides policymakers with deeper insights into the behavioral patterns of repeat offenders, allowing for the development of more targeted legal frameworks and monitoring measures to reduce the rate of drunk driving recidivism and the number of crashes and associated social costs caused by drunk driving, ultimately enhancing public safety.

In terms of academic contribution, this research expands the current scope of studies on drunk driving behavior by incorporating the PWM. Through a deeper analysis of the behavioral patterns and psychological motivations of repeat offenders, this study offers rich data and theoretical support for the fields of behavioral science, psychology, and criminology. Future research could build on the findings of this research to further explore other factors influencing drunk driving recidivism, as well as to design more effective behavior correction and intervention models, contributing to a more comprehensive understanding of drunk driving behavior within the academic community.

## 2. Theoretical Framework and Hypotheses Developments

### 2.1. The Prototype Willingness Model

The PWM proposes two pathways for behavioral performance ([35]): the reasoned path and the social reactive path (Figure 1). In the reasoned path, attitudes and subjective norms predict intentions, which in turn predict actual behavior, similarly to the theory of planned behavior (TPB), but without including perceived behavioral control (PBC). The social reactive path includes additional components beyond those in the TPB, specifically prototypes and willingness. Prototypes are images of a typical person who engages in the target behavior, characterized by the perceived similarity to and favorability of these prototypes to the individual. These prototypes influence behavioral willingness, which is the tendency to perform a behavior when the opportunity arises. The utility of the PWM has been demonstrated in various road user behaviors. Elliott and his colleagues compared the TPB and the PWM in the context of speeding behavior, and found that willingness is a stronger predictor of driver behavior than intention ([19]).

In the study of drunk driving behavior, the PWM provides a clearer and more direct explanation, particularly in analyzing the role of impulsivity and social influences. Drunk driving behavior is often driven by momentary impulses and immediate social reactions, which are not always based on rational thought. Therefore, the PWM has a stronger advantage in capturing these irrational and impulsive behaviors. For behaviors like drunk driving, which are often dominated by momentary impulses and situational factors, the PWM offers a more precise analytical framework. Drunk driving behavior is influenced by environmental pressures, social image, and immediate behavioral intentions, which are all core factors that the PWM emphasizes. Therefore, the PWM is better suited to explaining the immediate behavioral responses of drunk drivers, rather than solely focusing on rational behavioral intentions.

This study, based on the prototype willingness model, proposes a model to explain the intention, willingness, and recidivism behavior of drunk drivers. The hypotheses are shown, as follows:

**Hypothesis 1** **(H1).**
*The attitudes of drunk drivers have a significant influence on their intention to engage in drunk driving.*


**Hypothesis 2** **(H2).**The attitudes of drunk drivers have a significant influence on their willingness to engage in drunk driving.

**Hypothesis 3** **(H3).**
*The subjective norms of drunk drivers have a significant influence on their intention to engage in drunk driving.*


**Hypothesis 4** **(H4).**
*The subjective norms of drunk drivers have a significant influence on their willingness to engage in drunk driving.*


**Hypothesis 5** **(H5).**
*The prototype of drunk drivers has a significant influence on their willingness to engage in drunk driving.*


**Hypothesis 6** **(H6).**
*The willingness of drunk drivers has a significant influence on their intention to engage in drunk driving.*


**Hypothesis 7** **(H7).**
*The intention to engage in drunk driving of the drunk drivers has a significant influence on their actual drunk driving behavior.*


**Hypothesis 8** **(H8).**
*The willingness to engage in drunk driving of the drunk drivers has a significant influence on their actual drunk driving behavior.*


### 2.2. Additional Factors

In many countries, traffic regulations commonly use the threat of legal sanctions as a key strategy to prevent drunk driving and reduce its occurrence. There are also studies that have attempted to explore the impact of these deterrence measures on drivers’ drunk driving behavior. The research suggests that potential offenders’ perception of the likelihood of punishment plays an important role in their rational decision-making process when considering driving under the influence ([65]). Deterrence-related variables were introduced to measure the certainty of legal sanctions against drunk driving and to assess the perceived impact of random breath testing operations. These variables include both legal deterrence factors and non-legal deterrence factors ([31]). However, several important questions remain, including: (a) how effective legal deterrence is in influencing the intention, willingness, and behavior of drunk driving; (b) the impact of alcohol consumption on the decision to drive after drinking; (c) the effectiveness of legal awareness and education campaigns; and (d) the impact of legal penalties on reducing repeat drunk driving offenses.

Based on the literature review, it was found that integrating the PWM with the TPB model is beneficial ([28]; [47]). Therefore, we incorporated the TPB element into the structure of the PWM. According to the TPB, attitude, subjective norms, and PBC collectively influence an individual’s behavioral intentions and actions ([2]). PBC refers to a person’s beliefs and judgment about their ability to perform a specific behavior. It directly impacts both behavioral intention and behavior, and can also indirectly affect behavior, through its influence on intention. Based on these insights, we propose the following hypotheses.

**Hypothesis 9** **(H9).**
*The deterrence of drunk drivers has a significant influence on their intention to engage in drunk driving.*


**Hypothesis 10** **(H10).**
*The deterrence of drunk drivers has a significant influence on their willingness to engage in drunk driving.*


**Hypothesis 11** **(H11).**
*The deterrence of drunk drivers has a significant influence on their actual drunk driving behavior.*


**Hypothesis 12** **(H12).**
*The perceived behavioral control of drunk drivers has a significant influence on their intention to engage in drunk driving.*


**Hypothesis 13** **(H13).**
*The perceived behavioral control of drunk drivers has a significant influence on their actual drunk driving behavior.*


By expanding the original theoretical framework through the introduction of additional factors, this approach offers a more refined perspective that accounts for variables which are not covered in traditional models. The integration of these factors provides a more comprehensive understanding of the behavioral mechanisms under investigation, facilitating a deeper exploration of the interactions between existing constructs and newly identified influencing factors.

## 3. Methods

### 3.1. Participants and Procedures

The study was conducted from July to September 2024 in Taiwan, specifically at local motor vehicle offices. The participants were citizens who were penalized for drunk driving and were attending road traffic safety classes at these offices. To ensure the study’s rigor, the process began with the researchers informing the motor vehicle offices via official letters about the planned survey schedule, location, and target participants. Each local office then confirmed the feasible survey dates, and the surveys were conducted on-site, accordingly.

Upon receiving approval from the motor vehicle offices, the questionnaire was reviewed to ensure its appropriateness and to avoid any potential negative impact on the participants attending the classes. Participants were informed that the survey was solely for academic research purposes and would take approximately 20 to 30 min to complete. The questionnaire was administered using a guided reading approach, where the survey administrator read each question aloud one by one. Participants were asked to complete each question immediately after hearing it before moving on to the next question.

The questionnaire was divided into two parts. In the first part, participants provided their basic information, including their gender, age, marital status, highest level of education, and the amount of time since obtaining their driver’s license. The second part involved answering the survey questions. A total of 1156 questionnaires were collected, and after excluding incomplete or invalid responses, 855 valid questionnaires were analyzed.

Although the study participants were individuals who had been penalized for drunk driving, we strictly adhered to the relevant ethical guidelines throughout the research process. All participants underwent an informed consent procedure and participated voluntarily, with nobody being coerced into completing the survey. Regarding data privacy, we will ensure that all the collected data will remain anonymous, and will be used solely for academic research purposes. The questionnaire did not ask for any sensitive information that could identify individuals, and all responses will be kept confidential. Therefore, in both the recruitment and data handling processes, we have carefully considered and protected the ethical rights and privacy of the participants.

### 3.2. Questionnaire Design

This study developed a 40-item questionnaire to gather data from participants. The questionnaire included variables based on the prototype willingness model, such as attitude, subjective norm, prototype, drink driving intention, willingness, and behavior. It also integrated the perceived behavioral control component from the theory of planned behavior and elements of deterrence theory. Each item was measured using a five-point Likert scale. Additionally, the questionnaire collected demographic information from participants, including their gender, age, marital status, education level, occupation, monthly income, reasons for drink driving violations, and the duration since obtaining their driver’s license.

### 3.3. Measures

#### 3.3.1. Prototype Willingness Model

The PWM consists of six components: attitude, subjective norms, prototype, drink driving intention, drink driving willingness, and drink driving behavior. These components were measured according to real-life scenarios of drink drivers using a five-point Likert scale, ranging from one (strongly disagree) to five (strongly agree). The reliability of each measure was evaluated using Cronbach’s α, with α > 0.7 indicating acceptable internal consistency ([54]). All the measurement items are provided in detail in Table 2.

#### 3.3.2. Additional Factors

The additional factors consist of two components: perceived behavioral control and deterrence variables. These components were measured according to real-life scenarios of drink drivers using a five-point Likert scale, from one (strongly disagree) to five (strongly agree). The reliability of each measure was evaluated using Cronbach’s α, with α > 0.7 indicating acceptable internal consistency ([54]). All the measurement items are provided in detail in Table 3.

This study employed partial least squares structural equation modeling (PLS-SEM) for data analysis. The primary objective was to develop a framework based on theoretical foundations, and introduce new constructs to predict and explore the research variables. Given the need for overall model fit assessment, PLS-SEM is particularly suitable for tasks that emphasize variable explanation and prediction ([76]). PLS-SEM offers advantages, such as a reduced parameter estimation bias and an enhanced predictive analytical structure ([73]). In recent years, researchers have increasingly adopted PLS-SEM as an analytical approach to test structural models ([57]). Renowned for its flexibility and suitability for exploratory research ([38]), PLS-SEM is considered the optimal choice for analyzing complex structural models ([58]).

## 4. Data Analysis and Results

### 4.1. Descriptive Statistical Analysis

This study uses IBM SPSS Statistics 26 software to perform descriptive statistical analysis. Table 4 presents the demographic characteristics of the participants. Of the total respondents, 756 (88.4%) were male, and 99 (11.6%) were female. The largest age group was 46–55 years, comprising 223 participants (26.1%), followed closely by the 36–45 age group, with 222 participants (26%). Regarding educational background, the majority of participants, 517 individuals (60.4%), had completed high school or vocational education. A smaller proportion, 17.9%, held a university degree or higher, while 21.5% had not completed high school. In terms of driving experience, 619 participants (72.4%) had held their driver’s license for over 8 years, 54 participants had held it for 5–8 years, 69 participants for 2–5 years, and 113 participants for 0–2 years.

### 4.2. Measurement Model Testing

The study utilized SmartPLS 4 software to calculate the Cronbach’s alpha, factor loadings, rho_a, composite reliability (CR), and average variance extracted (AVE) for each construct, as shown in Table 5. In this study, the Cronbach’s alpha, rho_a, and CR values for each construct all exceed 0.7, indicating good reliability across all constructs ([5]; [18]; [62]). Previous research suggests that factor loadings should be above 0.5. In this study, all observed variables had factor loadings greater than 0.5, indicating that the measurement indicators possess good reliability. The AVE is used to assess convergent validity, with the general criterion being an AVE greater than 0.5. All indicators in this study meet this standard, demonstrating that the constructs possess good convergent validity.

This study used the heterotrait–monotrait ratio (HTMT) to assess discriminant validity. According to the HTMT criterion, discriminant validity is considered satisfactory when the HTMT value is less than 0.9, indicating good discriminant validity ([40]). Previous research has shown that the HTMT provides a more robust measure of discriminant validity compared to the Fornell–Larcker criterion ([41]). The results of this study demonstrate good overall discriminant validity, as shown in Table 6.

During the model evaluation phase, this study employed indicators, such as the coefficient of determination (R^2^) and predictive relevance (Q^2^), as criteria for assessing the model. The results showed that attitude, subjective norm, perceived behavioral control, deterrence and behavioral willingness explained 66.9% of the variance in drunk driving intention. Attitude, subjective norm, prototype, and deterrence explained 51.4% of the variance in drunk driving willingness. Similarly, perceived behavioral control, drunk driving willingness, drunk driving intention, and deterrence explained 41.9% of the variance in drunk driving behavior. All values are over 0.2, showing a satisfactory predictive power ([12]). To assess the predictive relevance, a blindfolding analysis was conducted, and the Q^2^ results indicated that the model demonstrated strong predictive relevance, with Q^2^ values for drunk driving intention (0.550) and willingness (0.453) both exceeding the cutoff of 0.35 ([44]).

Additionally, this study used the standardized root mean square residual (SRMR) and the normed fit index (NFI) to evaluate the fit of the structural equation model. The SRMR value was 0.072, which is below the threshold of 0.10, and the NFI value was 0.793, with values closer to one indicating a better model fit ([16]). These fit indices suggest that the hypothesized model proposed in this study performs well in explaining the data structure.

### 4.3. Hypotheses Testing

This study explores the factors influencing the intention, willingness, and behavior related to recidivism in drunk driving. The study applied a bootstrapping procedure (with 5000 bootstrap resamples), and collected a valid sample size of 855, to determine the significance of the paths in the structural model. Table 7 provides detailed structural estimates and hypothesis testing results. Among the 13 hypotheses, 11 were supported at the 0.05 significance level, while the remaining 2 hypotheses were not supported. According to the collinearity statistics, the variance inflation factor (VIF) values ranged from 1.210 to 2.897, all of which are below the recommended threshold of 5 ([39]), confirming that there is no multicollinearity issue in the estimated model of this study.

We also integrated the hypothesis testing results, along with the R-squared values and path coefficients of the relevant constructs in this study, into Figure 2 for visual representation.

## 5. Discussion

This study aims to apply the prototype willingness model to explore the predictive relationships among various factors influencing the reoffending intention, willingness, and behavior of drunk drivers who have already experienced being penalized for drunk driving. By utilizing a theoretical exploration approach, this study seeks to develop an integrated model combining multiple theories to identify a more comprehensive framework for explaining recidivism in drunk driving behavior. The study primarily employs the prototype willingness model, while incorporating the perceived behavioral control construct from the theory of planned behavior and elements from deterrence theory to achieve a completer and more suitable model for understanding this high-risk behavior. Previous research has mainly focused on examining drivers’ reoffending intentions, willingness, and behaviors, but there is currently no literature that specifically analyzes the recidivism behavior of drivers with prior drunk driving experience. Therefore, this study is highly indicative and potentially groundbreaking in this field.

This study is the first to apply the prototype willingness model (PWM) in interaction with the theory of planned behavior (TPB) to specifically understand the willingness to drink and drive among individuals who have previously engaged in drunk driving behavior. The findings of this study may hold significant implications for stakeholders involved in drunk driving education and road safety prevention programs. These programs need to focus on factors such as drivers’ attitudes, subjective norms, prototype, perceived behavioral control and deterrence.

This study hypothesized that attitudes, subjective norms, and behavioral willingness significantly impact behavioral intentions (H1, H3, H6), a hypothesis supported by numerous studies ([6]; [23]; [56]). Attitudes and subjective norms are key constructs for predicting behavioral intentions ([1]). Attitude refers to an individual’s positive or negative evaluation of a particular behavior, while subjective norms reflect the perceived social pressure or expectations from important others. Previous research indicates that the more positive an individual’s attitude toward a behavior and the stronger the perceived social pressure or expectations from others, the higher the behavioral intention ([1]; [4]). This suggests that attitudes and subjective norms are critical psychological factors in forming behavioral intentions, and significantly influence whether individuals are inclined to engage in specific behaviors. (H1 and H6 were supported).

However, the study results found that subjective norms did not predict the intention to drink and drive, a conclusion also supported by some studies ([9]; [48]; [60]). Further research has shown that subjective norms have a weaker relationship with the intention to participate in screening ([13]) or seek mental health services among university students ([3]). Attitudes, subjective norms, and perceived behavioral control are expected to vary in predicting intentions according to different behaviors and situations ([1]). For drunk drivers with hazardous and harmful alcohol consumption, perceived social pressure was not a decisive factor for help-seeking intentions. (H3 was not supported).

The study also confirmed that attitudes and subjective norms significantly impact behavioral willingness (H2, H4). Attitudes and subjective norms are key determinants of behavioral willingness ([1]). Research has shown that, when individuals hold a more positive attitude toward a behavior and perceive positive pressure from society or significant others, their willingness to engage in that behavior significantly increases ([1]; [63]). Therefore, attitudes and subjective norms are considered crucial constructs influencing behavioral willingness, and can effectively predict whether an individual is willing to perform a specific behavior. The influence of attitudes on willingness aligns with previous research that has assessed young drivers’ willingness to use mobile phones and engage in speeding behaviors ([19]; [61]). Thus, both the current and past research findings demonstrate that, when individuals hold favorable and positive attitudes toward risky behaviors, they are more inclined to engage in those risky behaviors. (H2 and H4 were supported).

Regarding the prototype component influencing the willingness toward drunk drive behavior (H5), the research results show a significant impact (estimated value = 0.504). This indicates that positive prototypes (including a favorable perception of the prototype and a higher perceived similarity to the prototype) are associated with a higher intention to engage in drunk drive behavior. These results suggest that individuals who perceive typical drunk drivers positively and see themselves as similar to these prototypes are more willing to engage in drunk drive behavior. These findings are consistent with previous studies, demonstrating that positive prototype perceptions are strong predictors of young individuals’ intentions to engage in risky behaviors, such as speeding ([59]). Overall, the results of these studies suggest that the prototype component effectively explains the extent and impact of young drivers’ intentions to engage in risky behaviors. (H5 was supported).

Our research results indicate that drunk driving intention and willingness are the main predictors of repeat drunk driving behavior (H7, H8), with the overall effect of willingness on drunk driving behavior significantly stronger than that of intention. Similar conclusions were reached by [66] ([66]), [19] ([19]), and [15] ([15]), who conducted studies on speeding among drivers and pedestrian violations, showing that behavioral willingness contributed more strongly to violating behavior than behavioral intention. Therefore, our study finds that the decision to drive under the influence is influenced more by the social reactive pathway than the reasoned pathway. This finding aligns with the conclusions of [37] ([37]) and [66] ([66]), who suggested that social reactive decision-making is more suitable for predicting risk behaviors. Thus, our study demonstrates that, when determining repeat drunk driving behavior, the social reactive decision-making aspect is more critical than reasoned decision-making. This discovery can be linked to driving behavior, indicating that socially reactive decision-making can respond effectively to changes in the traffic environment. (H7 and H8 were supported).

In this study, deterrence theory was introduced in the hypotheses (H9, H10, H11), positing its significant impact on drink-driving intentions, willingness, and behaviors. The findings confirmed that all these hypotheses were supported. Prior research has shown that the certainty and severity of punishment within deterrence theory significantly influence individuals’ behavioral intentions and willingness to comply with regulations. A higher certainty of punishment and perceived deterrence are associated with a reduction in the occurrence of violations, and individuals are more likely to adhere to regulations when the perceived level of deterrence is higher ([67]). Furthermore, the application of deterrence theory to traffic violations has demonstrated a significant impact on influencing behavior, such as reducing the likelihood of drunk driving ([42]), decreasing overall traffic violations by drivers ([75]), and mitigating speeding behavior ([20]; [27]). These findings highlight the effectiveness of deterrence theory in promoting compliance with traffic regulations and enhancing road safety. (H9, H10, and H11 were supported).

In this study, perceived behavioral control (PBC) was hypothesized to influence drink driving intentions and behaviors (H12, H13). This hypothesis aligns with the assumptions of the Theory of Planned Behavior (TPB) model and was incorporated into the current research framework. The findings indicated that PBC significantly affects drink driving intentions, consistent with previous research ([10]), demonstrating that participants confident in their ability to drive after drinking are indeed more likely to intend to engage in drink driving. (H12 was supported).

However, on the other hand, the results showed that PBC did not significantly impact actual drink driving behavior, aligning with previous studies showing similar findings ([8]; [11]; [46]; [53]). In examining the relationship between PBC and actual behavior, some research has found that PBC does not always significantly influence actual behavior. Factors such as past behavior, the stability of intentions, and familiarity with the behavior can moderate the effect of PBC. Specifically, when individuals are unfamiliar with the behavior or have lower past involvement, the impact of PBC on behavior tends to be weaker or even insignificant. For instance, PBC has been found to be a significant predictor of behavior only among those familiar with the behavior, suggesting that in unfamiliar contexts, PBC may not effectively reflect actual control capacity ([53]). (H13 was not supported).

## 6. Conclusions and Policy Implications

### 6.1. Conclusions

This study aims to analyze the impact of prototype willingness model (PWM) constructs, with the addition of theory of planned behavior (TPB) components, on recidivism behavior in drunk driving, and to establish a final model framework by exploring the relationships among the constructs. Drawing on previous research, relevant constructs were introduced to understand and explore the positive relationships between intentions, willingness, and actual behavior. The results indicated that drunk driving intentions are positively influenced by attitudes, perceived behavioral control, and willingness to engage in drunk driving behavior, while also being negatively affected by deterrence. Furthermore, behavioral willingness emerged as the most important factor influencing behavioral intentions. Willingness is positively influenced by attitudes and prototypes, but is negatively impacted by subjective norms and deterrence. Recidivism in drunk driving behavior is positively influenced by both intentions and willingness to engage in drunk driving, while deterrence exerts a negative effect. The primary factor influencing recidivism is behavioral willingness.

The findings of this study indicate that subjective norms do not have a significant impact on drunk driving intentions. This suggests that individuals do not appear to be influenced by the opinions or views of family, friends, or significant others when considering their intent to drive under the influence. Despite being aware that such behavior is not endorsed by their social circles, individuals may still exhibit intentions to engage in drunk driving due to perceived immediate needs or urgency.

In terms of limitations, this study focuses primarily on understanding the recidivism of drunk drivers. However, individuals with no prior experience of driving under the influence (first-time offenders) could also exhibit drunk driving behavior. Future research aiming for a broader understanding of drunk driving behavior may benefit from including non-offending individuals in the survey, to provide a more comprehensive perspective on perceived behavioral control and its role in influencing intentions. Future related research could further explore the mediating effects between intentions, willingness, and actual behavior, as well as the overall impact among the constructs, providing broader research insights. To increase the generalizability of the study, the sample could be expanded to other regions, or people without any experience of drunk driving could be included in the future. Some qualitative research tools could be incorporated in the future to further explore the psychological motivation of the respondents. We also suggest that future studies could compare participants from different cultural backgrounds, and further explore how individual traits (such as self-control, risk propensity, etc.) affect the formation of drunk driving intentions and their interaction with subjective norms.

### 6.2. Policy Implications

The study participants were a unique group, all of whom had been apprehended for drunk driving by law enforcement, and subsequently mandated to attend road safety education sessions organized by government regulatory agencies. Therefore, the developed research model can be utilized in government drunk driving prevention and education campaigns. The findings also allow for the assessment of the appropriateness of existing road safety course content. The study’s findings indicate that constructs such as behavioral willingness, intention to drink and drive, and deterrence significantly influence drunk driving behavior. As a result, regulatory agencies designing drunk driving prevention courses could incorporate these three constructs into the curriculum to enhance the material’s depth. Additionally, interviews with individuals convicted of drunk driving could provide valuable insights into how these factors interact. This approach could contribute further to efforts aimed at reducing drunk driving behavior and preventing recidivism.

The results of this study not only provide insights for future analysis on related topics, but can also be applied to policy analysis and improvements. The findings may serve as a reference for legislators and policymakers addressing issues related to drunk driving laws, guiding them to gradually review the appropriateness of current penalties based on the analysis of recidivism behavior. Furthermore, the findings can be continuously applied to explore psychological aspects related to drunk driving offenders, offering both practical and academic contributions to society.

In recent years, Taiwan has experienced a high incidence of drunk driving crashes, leading to severe traffic collisions and casualties. To effectively mitigate the risks associated with driving under the influence, promoting designated driver services is essential. Designated driver services provide a safe alternative for individuals to return home after consuming alcohol, which not only reduces the likelihood of crashes due to impaired driving, but also helps individuals avoid the legal consequences of drunk driving.

The promotion of designated driver services not only enhances public safety, but also raises awareness about responsible driving within society. Through targeted campaigns and education, it is crucial to increase public understanding of the benefits of designated driver services, encouraging more people to choose this option after drinking, rather than risking driving under the influence. By doing so, the rate of drunk driving-related crashes could be significantly reduced. Furthermore, the government should consider implementing supportive legislation, such as setting reasonable pricing for designated driving services, to encourage the broader use of this safer alternative. This approach could effectively lower Taiwan’s rate of repeat drunk driving offenses.

Many countries are currently implementing alcohol interlock policies, as alcohol interlocks can effectively prevent drunk driving behaviors and improve overall societal and traffic safety. Drivers subject to alcohol interlock measures must pass an alcohol test before starting their vehicles, making it a more effective tool in restricting drunk driving. This is particularly beneficial for high-risk drinkers and repeat offenders, as alcohol interlocks serve as a relatively effective measure to reduce recidivism. Previous studies have indicated that over 60% of drunk drivers are willing to install alcohol interlocks in exchange for the right to drive, but their willingness to pay is significantly lower than the government-approved product price. Personal attitudes towards alcohol interlocks influence respondents’ willingness to use and pay for them. By promoting road safety education programs or advertisements, the positive image of alcohol interlocks can be enhanced, leading to an increase in the overall number of installations ([45]).

Government agencies can adopt multiple policy measures to reduce drunk driving behavior and raise public awareness of traffic safety. First, we should strengthen the popularization of drunk driving risk education from an early age, by utilizing various media and promotional channels to emphasize the dangers of drunk driving and the legal consequences that must be faced. Additionally, public awareness campaigns should include the dissemination of relevant laws and penalties related to drunk driving.

Second, the government should encourage collaboration between social enterprises and private organizations to implement targeted educational activities for specific groups, such as repeat drunk drivers, young drivers, and professional drivers, to enhance their understanding of the risks associated with drunk driving.

Finally, there should be an increase in the enforcement of drunk driving laws, with improvements in the technology and methods used for detection, such as expanding the number of sobriety checkpoints and utilizing faster alcohol testing devices. At the same time, an effective legal punishment and disposal mechanism for drunk drivers should be established, to serve both as a deterrent and as a means of ensuring accountability.

## Figures and Tables

**Figure 1 behavsci-15-00048-f001:**
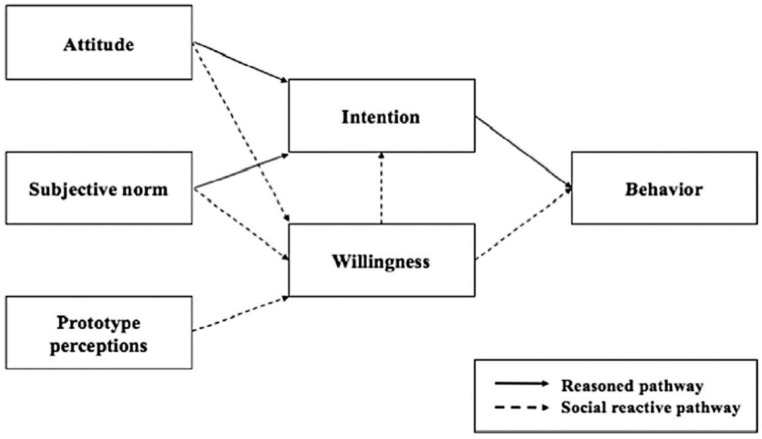
The prototype willingness model ([35]).

**Figure 2 behavsci-15-00048-f002:**
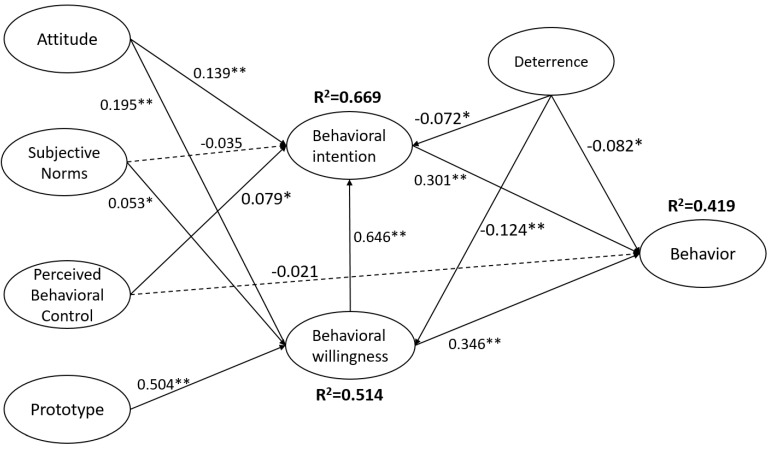
Path model shows the hypothesis relationships between PWM variables, perceived behavioral control, and deterrence variables. Note: Bold lines indicate significant paths. Dotted lines indicate non-significant paths; * *p* < 0.05, ** *p* < 0.001.

**Table 1 behavsci-15-00048-t001:** Statistics on drunk driving fatalities in Taiwan from 2009 to 2023.

**Year**	**2009**	**2010**	**2011**	**2012**	**2013**	**2014**	**2015**	**2016**
Drunk driving fatalities	397	419	439	376	245	169	142	102
**Year**	**2017**	**2018**	**2019**	**2020**	**2021**	**2022**	**2023**	
Drunk driving fatalities	87	100	149	151	163	140	137	

**Table 2 behavsci-15-00048-t002:** Measurement scale of PWM factors.

Constructs	Items	Content	Source
Attitude		1 = “strongly disagree” to 5 = “strongly agree”	([2]; [7]; [49]; [68]; [72])
	AT1	I believe driving after drinking is a wise decision.	
	AT2	I think driving after drinking is a good idea.	
	AT3	I consider driving after drinking to be the right choice.	
	AT4	Driving home after drinking at a party feels comfortable to me.	
Subjective Norms		1 = “strongly disagree” to 5 = “strongly agree”	([2]; [7]; [49]; [68]; [72])
	SN1	My friends do not want me to drive after drinking at late-night parties.	
	SN2	My family does not expect me to drive after drinking.	
	SN3	Most important people in my life would not approve of me driving after drinking at parties.	
Prototype		1 = “strongly disagree” to 5 = “strongly agree”	([7]; [49]; [66]; [68]; [72])
	PT1	I have a good impression of drink drivers of my age.	
	PT2	I think the behavior of drink drivers of my age is very mature.	
	PT3	I think a drink driver of my age is energetic.	
	PT4	I think drink drivers my age are independent thinkers.	
	PT5	I have similar characteristics with drink drivers of my age.	
	PT6	My behavior is similar to that of drink drivers of my age.	
	PT7	I can be compared with drink drivers of my age.	
	PT8	I am as confident as drink drivers of my age.	
Behavioral Intention		1 = “strongly disagree” to 5 = “strongly agree”	([43]; [49]; [66]; [72])
	BI1	I will continue to engage in drunk driving.	
	BI2	I will still drive after drinking at parties.	
	BI3	I will choose to drive after drinking at a friend’s house.	
	BI4	I am unable to avoid drunk driving.	
Behavioral Willingness		1 = “strongly disagree” to 5 = “strongly agree”	([7]; [28]; [49]; [66]; [68])
	BW1	I am still willing to drive after drinking.	
	BW2	I would drive after drinking when leaving gatherings with friends or family.	
	BW3	I expect to engage in drunk driving in the coming months.	
Behavior	BE1	I have driven drunk in the past 30 days. From 1 to 5 (strongly disagree to strongly agree).	([19]; [21]; [66])
	BE2	How often have you driven drunk in the past 30 days? (From 1 to 5: Never, Very Rarely, Rarely, Occasionally, Very Frequently).	
	BE3	How many times have you driven drunk in the past 30 days? (From 1 to 5: Never, Rarely, Sometimes, Often, Always).	

**Table 3 behavsci-15-00048-t003:** Measurement scale of additional factors.

Constructs	Items	Content	Source
Perceived Behavioral Control		1 = “strongly disagree” to 5 = “strongly agree”	([2]; [49]; [66]; [68]; [72])
	PBC1	I can easily control a vehicle and assess situations even after drinking.	
	PBC 2	I am confident in my driving abilities even after drinking.	
	PBC 3	Drinking does not affect my driving skills.	
Deterrence Variables		1 = “strongly disagree” to 5 = “strongly agree”	([30], [31]; [32]; [64])
	DV1	I think the chances of being caught for drunk driving are high.	
	DV2	If I drive drunk, I am likely to be caught.	
	DV3	Drunk driving would significantly impact my life.	
	DV4	The punishment for drunk driving would be severe for me.	
	DV5	After being caught for drunk driving, the time to go to court is short, and I would pay the price.	
	DV6	If I drive drunk, I will soon lose my license and become an unlicensed driver.	
	DV7	If I drive drunk, I worry about losing my friends’ respect.	
	DV8	If I were caught for drunk driving, I would feel ashamed if my friends found out.	
	DV9	I would feel guilty after drunk driving.	
	DV10	I would feel stupid for drunk driving.	
	DV11	If I drive drunk, I worry about getting injured or hurt.	
	DV12	I think drunk driving poses a serious risk to my health.	

**Table 4 behavsci-15-00048-t004:** Demographic characteristic of the respondents (N = 855).

Variables	Items	Frequency	Percentage
Gender	Male	756	88.4%
	Female	99	11.6%
Age	18–25	88	10.3%
	26–35	137	16%
	36–45	222	26%
	46–55	223	26.1%
	56–65	140	16.4%
	66–75	44	5.1%
	Over 76	1	0.1%
Marital Status	Single	337	39.4%
	Married	333	38.9%
	Divorce	165	19.3%
	Widowed	20	2.3%
Education Level	Primary School	31	3.6%
	Secondary School	153	17.9%
	High School	427	49.9%
	Junior College	90	10.5%
	Bachelor’s Degree	126	14.7%
	Master’s Degree	26	3%
	Doctorate	2	0.2%
License	0–2 years	113	13.2%
	2–5 years	69	8.1%
	5–8 years	54	6.3%
	Over 8 years	619	72.4%

**Table 5 behavsci-15-00048-t005:** Measurement scales, reliability, and validity analysis.

Constructs	Items	Cronbach’s α	Factor Loadings	Rho-A	CR	AVE
Attitude		0.945		0.947	0.961	0.86
	AT1		0.915			
	AT2		0.930			
	AT3		0.961			
	AT4		0.902			
Subjective norms		0.956		0.957	0.971	0.919
	SN1		0.951			
	SN2		0.963			
	SN3		0.962			
Prototype		0.914		0.944	0.927	0.617
	PT1		0.842			
	PT2		0.882			
	PT3		0.893			
	PT4		0.861			
	PT5		0.648			
	PT6		0.654			
	PT7		0.689			
	PT8		0.765			
Behavioral intention		0.931		0.936	0.952	0.831
	BI1		0.939			
	BI2		0.947			
	BI3		0.946			
	BI4		0.806			
Behavioral willingness		0.939		0.939	0.961	0.891
	BW1		0.954			
	BW2		0.951			
	BW3		0.926			
Behavior		0.864		0.902	0.915	0.782
	BE1		0.898			
	BE2		0.877			
	BE3		0.878			
Perceived behavioral control		0.956		0.958	0.972	0.92
	PBC1		0.955			
	PBC 2		0.965			
	PBC 3		0.957			
Deterrence variables		0.947		0.954	0.953	0.63
	DV1		0.757			
	DV2		0.749			
	DV3		0.816			
	DV4		0.769			
	DV5		0.772			
	DV6		0.699			
	DV7		0.761			
	DV8		0.790			
	DV9		0.858			
	DV10		0.860			
	DV11		0.850			
	DV12		0.829			

**Table 6 behavsci-15-00048-t006:** Discriminant validity: heterotrait–monotrait ratio (HTMT).

	AT	BI	BW	BE	DV	PBC	PT	SN
AT								
BI	0.604							
BW	0.574	0.847						
BE	0.457	0.653	0.660					
DV	0.475	0.485	0.470	0.392				
PBC	0.463	0.452	0.432	0.294	0.300			
PT	0.547	0.692	0.682	0.511	0.422	0.517		
SN	0.293	0.280	0.261	0.264	0.429	0.018	0.164	

**Table 7 behavsci-15-00048-t007:** Path coefficients and the results of the significance tests.

Hypothesis	Path	Estimate	t Statistics	*p*-Values	VIF	f^2^	95%CI	Results
H1	AT-BI	0.139	3.721	0.000	1.680	0.035	[0.074, 0.218]	Supported
H2	AT-BW	0.195	4.400	0.000	1.582	0.049	[0.115, 0.289]	Supported
H3	SN-BI	−0.035	1.299	0.194	1.254	0.003	[−0.09, 0.015]	Not supported
H4	SN-BW	−0.053	2.132	0.033	1.210	0.005	[−0.103, −0.005]	Supported
H5	PT-BW	0.504	9.981	0.000	1.521	0.344	[0.401, 0.598]	Supported
H6	BW-BI	0.646	15.845	0.000	1.604	0.785	[0.559, 0.722]	Supported
H7	BI-BE	0.301	4.398	0.000	2.897	0.054	[0.169, 0.434]	Supported
H8	BW-BE	0.346	4.691	0.000	2.816	0.073	[0.204, 0.493]	Supported
H9	DV-BI	−0.072	2.947	0.003	1.516	0.010	[−0.122, −0.026]	Supported
H10	DV-BW	−0.124	3.611	0.000	1.493	0.021	[−0.195, −0.06]	Supported
H11	DV-BE	−0.082	2.315	0.021	1.319	0.009	[−0.154, −0.016]	Supported
H12	PBC-BI	0.079	3.477	0.001	1.363	0.014	[0.034, 0.124]	Supported
H13	PBC-BE	−0.021	0.768	0.442	1.257	0.001	[−0.075, 0.035]	Not supported

## Data Availability

Data will be provided upon request.

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
