# Peer review of "Determinants for Drunk Driving Recidivism—An Application of the Integrated Prototype Willingness Model"

_behavsci, 2025, doi:10.3390/bs15010048_

Round 1
Reviewer 1 Report
Comments and Suggestions for Authors
First of all, I would like to recognize your research, which delved into the psychological and social factors of recidivist DUI offenders through the prototype willingness model, a study that was very innovative and provided many valuable insights. The hypotheses you proposed were rigorously analyzed and the results of the questionnaire were very rich, especially in terms of the influence of intention and willingness on behavior.
However, there are a few areas where I feel that you could have gone a little deeper or expanded, and I hope that my suggestions will help you to further enhance the depth of your research. For example, the scope of your current sample focuses on the group of drunk drivers in Taiwan, and the results of the study are highly applicable to a specific region, but if you have the opportunity in the future, perhaps you can consider expanding the sample to other regions, or even include people who have no experience of drunk driving, which can increase the generalizability of the study.
Also, I noticed that you used mostly quantitative research methods, which are very effective in measuring attitudes, intentions, etc. However, if you can incorporate some qualitative research tools, such as interviews to further explore the psychological motivation of the respondents, it may make the results richer and more three-dimensional. Of course, this will also need to be adjusted appropriately in light of the specific goals and time and effort of your research.
Regarding the point that subjective norms do not play a significant role in predicting DUI intentions, it might be useful to further explore whether there is any influence of cultural background or individual traits, and why the role of social pressures is diminished in certain contexts, which might provide more directions for subsequent research.
Overall, I really appreciate your research ideas and efforts, and these suggestions are for reference only, and I hope they can shed a little light on your research. I look forward to seeing your further results!
Reviewer 2 Report
Comments and Suggestions for Authors
This study used the prototype willingness model to evaluate the factors influencing the behavioral intention to drink and drive. Overall, the study used a consistent methodology and provides valuable insights on the concept; however, there are several parts that can be improved:
Abstract (line 14-16):
Add more information about your sample in the abstract (e.g., number of participants).
Introduction (line 131): the last paragraph of the introduction section regarding the structure of the paper is not needed.
Introduction: I believe the context of the study has not been introduced completely for readers. For instance, what are the countermeasures in place in the study country, including the rules, public education of the issue, and any previous campaigns on drink and drive can be presented.
Theoretical Framework (line 149-151):
The rationale for choosing PWM model should be more consolidated. While the introduction discussed some previous studies on the strengths of PWM model, I believe more comprehensive justification could be provided. I believe that some recent studies also showed the superiority of the TPB and TPB-PWM integration model compared to PWM in other behavioural contexts. Therefore, a thorough argument comparing the behavioural intention models could improve the manuscript. You can use the following references as a start point: 10.1016/j.trf.2024.09.003 and 10.1016/j.tra.2022.103565 and 10.1016/j.aap.2017.09.011
Method:
Considering that this study has recruited individuals penalized for drunk driving, what were the ethical considerations for recruitment and privacy of the data. I believe it is worth to have a paragraph in the method section elaborating on the ethical considerations of this study.
Results (line 336):
The NFI value is not satisfactory for showing the model’s goodness of fit. I believe you should emphasize other reliability and validity measures that shows the appropriateness of the model and acknowledge that the NFI value is below the recommended threshold.
Results: There is previous evidence showing that the Fornell–Larcker criterion might not provide a good measure of discriminant validity of the model. I believe the HTMT ratio is enough.
Results: Table 8 looks redundant. You mention briefly in the text, however, it is already in the figure. Table 9 can also be in the text instead of a table.
Discussion (line 464): The last paragraph of the discussion looks disconnected. Please reconsider the flow of this section.
Conclusions (line 469):
The limitations of this study have not been stated completely. I believe more can be provided in this paragraph.
Policy implications:
I believe more elaboration on providing enough information about the risks of the behaviour for drivers could be seen in this subsection.
Reviewer 3 Report
Comments and Suggestions for Authors
The study of the current manuscript explores the psychological aspects of drunk drivers in order to highlight a deeper understanding of their mental and cognitive characteristics. The authors' ambition is to reveal how various factors influence behavioral intentions and willingness to drive under the influence of alcohol and the potential impact they have on recidivism.
It is known that investigating the intentions, willingness and recidivism of drunk driving offenders is crucial for the prevention and control of drunk driving and the authors' innovation and scientific contribution to the international literature is the integration of PWM in the study of drunk driving behavior.
The authors, using sophisticated analyses (e.g. PLS-SEM) and based on well-known and used theories (e.g. TPB) proceeded to a (perhaps) exhaustive analysis of the behavioral patterns and psychological motivations of repeat offenders. For my part, I would like to suggest some small improvements such as, for example, the use of the terms incidents or crashes instead of the unfortunate term accidents (the issue of road collision is a matter of driver responsibility and not of lack, fate, etc.).
In addition, I would like to see an elegant argument regarding the increased number of questionnaires rejected by the researchers and the possible impact on their results. In conclusion, I believe that this work would be of interest to the readers of your journal and I would be very happy to see it published.
Round 2
Reviewer 2 Report
Comments and Suggestions for Authors
The authors have made commendable efforts to address the previous feedback, and the manuscript has improved significantly. However, a few concerns remain:
Response to Comment #5: The response to my previous comment regarding the theoretical framework remains insufficient. Several relevant articles, both in terms of modeling and subject matter, have not been addressed. This issue can also affect the manuscript arguments and discussion.
Response to Comment #6: While the addition of a section on ethics and data privacy in the methodology is insightful, does this imply that ethics approval was not obtained from a third party, such as a university or an independent committee? Clarification on this matter is necessary, as ethical approval is typically required for survey-based studies.
Response to Comment #8: The authors simply acknowledged this comment with a thank-you note, but their actions regarding this feedback remain unclear. A more explicit explanation of how this comment has been addressed is necessary.
Repetition of Sources: Some references are duplicated within the manuscript, such as Gibbons et al. (2006). This issue should be corrected to maintain the clarity and professionalism of the reference list.
Round 3
Reviewer 2 Report
Comments and Suggestions for Authors
The authors have responded thoughtfully to the previous feedback, and the manuscript shows significant improvement. While there is still some room for enhancement, I recommend accepting the manuscript to avoid unnecessary delays in publication.
Author Response
We appreciate your positive feedback.